# Antioxidant vs. Prooxidant Properties of the Flavonoid, Kaempferol, in the Presence of Cu(II) Ions: A ROS-Scavenging Activity, Fenton Reaction and DNA Damage Study

**DOI:** 10.3390/ijms22041619

**Published:** 2021-02-05

**Authors:** Miriama Simunkova, Zuzana Barbierikova, Klaudia Jomova, Lenka Hudecova, Peter Lauro, Saleh H. Alwasel, Ibrahim Alhazza, Christopher J. Rhodes, Marian Valko

**Affiliations:** 1Institute of Physical Chemistry and Chemical Physics, Faculty of Chemical and Food Technology, Slovak University of Technology, 812 37 Bratislava, Slovakia; miriama.simunkova@stuba.sk (M.S.); zuzana.barbierikova@stuba.sk (Z.B.); marian.valko@stuba.sk (M.V.); 2Department of Chemistry, Faculty of Natural Sciences, Constantine the Philosopher University in Nitra, 949 74 Nitra, Slovakia; kjomova@ukf.sk (K.J.); lhudecova@ukf.sk (L.H.); plauro@ukf.sk (P.L.); 3Zoology Department, College of Science, King Saud University, Riyadh 12372, Saudi Arabia; salwasel@ksu.edu.sa (S.H.A.); imalhazza@hotmail.com (I.A.); 4Fresh-Lands Environmental Actions, Reading, Berkshire RG4 5BE, UK

**Keywords:** kaempferol, flavonoid, copper, antioxidant, prooxidant, spectroscopy, viscometry, gel-electrophoresis

## Abstract

Kaempferol is a flavonoid that occurs in tea and in many vegetables and fruits, including broccoli, cabbage, beans, grapes, apples, and strawberries. The efficacy of Kaempferol has been demonstrated in the treatment of breast, esophageal, cervical, ovarian, and liver cancers and leukemia, which very likely arises from its prooxidant properties and the activation of pro-apoptotic pathways. Indeed, this matter has already been the focus of a number of published studies and reviews. The aim of the present study was to elucidate the antioxidant vs. prooxidant properties of flavonoids in the presence of the redox-active metal, copper (II) ion, by means of the Fenton reaction. The specific motivation of this work is that, since an increased level of Cu(II) ions is known to be associated with many disease states such as neurological conditions (Alzheimer’s disease) and cancer, any interaction between these ions and flavonoids might affect the outcome of therapeutic uses of the latter. The structure of the Cu-kaempferol complex in DMSO was investigated by means of low temperature EPR spectroscopy, which confirmed the existence of at least two distinct coordination environments around the copper (II) ion. UV vis-spectra of kaempferol and its Cu(II) complex in DMSO revealed an interaction between the 5-OH (A ring) group and the 4-CO (C ring) group of kaempferol with Cu(II) ions. An ABTS assay confirmed that kaempferol acted as an effective radical scavenger, and that this effect was further enhanced in the form of the Cu(II)-kaempferol complex. Quantitative EPR spin trapping experiments, using DMPO as the spin trap, confirmed suppression of the formation of a mixture of hydroxyl, superoxide, and methyl radicals, in a Fenton reaction system, upon coordination of kaempferol to the redox-active Cu(II) ions, by 80% with respect to the free Cu(II) ions. A viscometric study revealed a better DNA-intercalating ability of the Cu-kaempferol complex than for free kaempferol, essential for conferring anticancer activity of these substances. The results of the viscometric measurements were compared with those from a DNA damage study of Cu-kaempferol complexes in a Fenton reaction system, using gel electrophoresis. At low concentrations of kaempferol (Cu–kaempferol ratios of 1:1 and 1:2), a very weak protective effect on DNA was noted, whereas when kaempferol was present in excess, a significant DNA-protective effect was found. This can be explained if the weakly intercalated kaempferol molecules present at the surface of DNA provide protection against attack by ROS that originate from the Fenton reaction involving intercalated Cu(II)-kaempferol complexes. Following the application of ROS scavengers, L-histidine, DMSO, and SOD, gel electrophoresis confirmed the formation of singlet oxygen, hydroxyl radicals, and superoxide radical anions, respectively. We propose that the prooxidant properties of Cu-kaempferol complexes may provide anticancer activity of these substances. When present in excess, kaempferol displays antioxidant properties under Cu-Fenton conditions. This suggests that kaempferol might prove a suitable candidate for the prevention or treatment of oxidative stress related medical conditions that involve a disturbed metabolism of redox metals such as copper, for example, Menkes disease, and neurological disorders, including Alzheimer’s disease. For the potential use of kaempferol in clinical practice, it will be necessary to optimize the dose size and critical age of the patient so that this flavonoid may be beneficial as a preventive drug against cancer and neurological disorders.

## 1. Introduction

In addition to their physiological roles in plants, flavonoids may function as important antioxidants, as obtained from the human diet [1]. Beneficial roles for flavonoids in the prevention of human diseases are related either to their direct antioxidant or, indeed, mild prooxidant activities [2]. Such activities can be manifested by an ability to scavenge free radicals, chelate redox-active metal ions, activate antioxidant enzymes, activate the synthesis of low molecular mass antioxidants such as glutathione, regenerate vitamin E from its radical form, alleviate nitric oxide-induced oxidative stress, inhibit oxidases, and other beneficial modes of action [3].

The antioxidant or prooxidant properties of flavonoids depend predominantly on the number and positions of substituent hydroxyl groups, along with their related capacity to chelate redox-active metal ions such as those from copper and iron [3,4]. Another important structural feature that affects the antioxidant/prooxidant properties of flavonoids is the planarity of the molecular skeleton and extent of π-electron delocalization. The flavonoid B ring appears to be a biologically important functionality in ROS scavenging, which operates via hydrogen-atom and/or electron transfer to hydroxyl, peroxyl, and other radical species. Such transfers result in the formation of resonance stabilized flavonoid radicals.

Previous studies have shown that, in the presence of metals, the behavior of flavonoids is very complex and the basis of their antioxidant activity, as a result of ROS scavenging or metal chelation is still controversial [4]. A more pronounced antioxidant activity for Cu(II)-quercetin complexes than for quercetin alone has been observed [5,6]. On the other hand, many in vitro experiments have confirmed that, when in combination with redox-active metals, flavonoids show prooxidant effects that lead to DNA fragmentation [7,8]. Our group [4,9] and others [10] have suggested that the prooxidant effect of polyphenols could be a significant part of the mechanism for their anti-tumor activity. Flavonoids in copper-rich cells can induce the formation of ROS through redox cycling and subsequent DNA fragmentation, and thereby induce apoptosis of tumor cells [11,12]. Tan et al. [13] studied Cu(II)-flavonoid complexes and suggested that it is not only the formation of ROS leading to oxidative DNA damage, but also specific interactions with DNA, that must be part of the anti-tumor mechanism of these complexes. The ability of their molecules to interact with DNA makes flavonoids interesting in terms of their potential use for cancer treatment, in which the target molecule is primarily that of DNA. Since those substances with an ability to intercalate into DNA are of particular importance, it is necessary to study how natural flavonoids interact with DNA under in vitro conditions.

In the present study, we have evaluated the antioxidant/prooxidant properties of kaempferol (Figure 1) in the presence of Cu(II) ions, under the conditions of a model Fenton type reaction system. The beneficial health effect of kaempferol has been demonstrated in the treatment of breast, esophageal, cervical, ovarian, and liver cancers and leukemia, which very likely arises from its prooxidant properties and the activation of pro-apoptotic pathways [14,15,16,17,18,19,20,21,22]. In our recent work, we studied the interaction of Cu(II) ions with quercetin and determined how this impacted on the antioxidant and prooxidant properties of the flavonoid [4]. Since quercetin differs from kaempferol only in having one more -OH group on ring B (at the 3-position), it is rational to compare these two flavonoids in terms of their complexation properties with Cu(II) ions and corresponding antioxidant and prooxidant behaviour. Our particular aim was to study the antioxidant vs. prooxidant properties of kaempferol and its Cu(II) complexes in regard to their interaction with DNA, and ROS formation via the Fenton reaction. 

## 2. Results and Discussion 

The overall aim of this work was to study the effect of cupric ions on the antioxidant/prooxidant properties of kaempferol. Firstly, we discuss the interaction between Cu(II) and kaempferol by means of EPR and UV-vis spectroscopy, in order to ascertain the bonding mode between a given cupric ion and kaempferol molecules. The latter study was supplemented by determinations of the radical scavenging activity of kaempferol, EPR spin trapping experiments using a model Fenton reaction, and measurements of the interaction of the Cu(II)-kaempferol complex with DNA using gel electrophoresis and viscometric methods.

### 2.1. EPR Spectroscopy

The EPR spectrum of the Cu(II)-kaempferol (1:1) complex in frozen DMSO solution is shown in Figure 2. Since the spectrum is significantly different from that of the CuCl_2_ standard in DMSO (spectrum not shown), an interaction having occurred between kaempferol and the Cu(II) ion is confirmed. The high value of g_‖_ and the presence of two separate hyperfine splitting patterns indicates that the Cu(II) ions can adopt at least two distinct distorted coordination environments (Table 1) [23]. We propose that the remaining binding sites around the Cu(II) ion and the kaempferol molecule are most probably occupied by DMSO molecules. In order to obtain additional information, to that from the EPR measurements, a complementary study was made using UV-vis spectroscopy. 

### 2.2. UV-Vis Spectroscopy

UV-vis spectra of kaempferol and its copper complexes (1:1 and 1:2) in DMSO and in methanol are shown in Figure 3 and Figure 4, respectively. 

In summary, the UV-vis spectra of flavonoids are characterized by π → π* transitions at about 240–290 nm, which are associated with the rings A and C [24]. These transitions are usually denoted as bands II and belong to benzoyl system of the skeleton. Transitions occurring at about 300–415 nm belong to the B-ring, which is conjugated with the C-ring carbonyl (cinnamoyl system) and are referred to as band I [25]. Formation of a metal-flavonoid complex causes a marked red (bathochromic) shift of band I by ca 100 nm toward longer wavelengths. Such a shift can be explained by the formation of a more extended conjugated system following the formation of a new ring by chelation of the Cu(II) ion [26]. 

UV-vis spectra of free kaempferol and the Cu(II)-kaempferol complexes in DMSO show moderate differences (Figure 3). Upon the formation of the Cu(II)-kaempferol complex, a new band appears at 290 nm, which indicates the interaction of the 5-OH group on ring A and the 4-CO group on ring C of kaempferol with Cu(II) ions [27,28,29]. The absence of any significant changes in the 300–415 nm region of the UV-vis spectrum indicates a negligible or very weak interaction between the Cu(II) ions and 4-CO and 3 -OH groups of the kaempferol ring C [30]. 

In order to ascertain how the properties of different solvent molecules affect the metal-flavonoid interaction, UV-vis spectra were recorded from the copper(II)-kaempferol complexes in methanol, as are shown in Figure 4. The spectra show a marked shift, of about 65 nm, in band I indicating a tight interaction of the 4-CO and 3-OH groups from the kaempferol ring C with Cu(II) ions. A minor shift of the kaempferol band at 264 nm, by 5 nm, towards higher wavelengths indicates a weak interaction of the 5-OH group on ring A and of the 4-CO group on ring C with Cu(II) ions. The marked differences in copper(II)-kaempferol binding modes between DMSO and methanol, as documented by UV-vis spectra, are due to the different strengths of the interactions between different solvent molecules and Cu(II) ions. Since the donor numbers of DMSO and methanol are 40 and 19, respectively, it is expected that the interaction between Cu(II) and DMSO solvent molecules will be significantly stronger than that between Cu(II) and methanol solvent molecules [31]. This results in an adoption of different binding modes between Cu(II) and kaempferol, and generally any flavonoid, according to the nature of the solvent employed, which is manifested in the according UV-vis spectra. Although the spectra recorded in methanol were better resolved, the use of DMSO is substantially a better choice in terms of the biological relevance of the results, and indeed, DMSO is often employed as a model solvent in biological studies, while methanol is a toxic substance.

### 2.3. Radical Scavenging Activity of Kaempferol and Cu(II)-Kaempferol (1:1 and 1:2) Complexes 

The radical scavenging activity of kaempferol, both in the absence and in the presence of Cu(II) ions, was studied using the ABTS assay (Appendix A). In general, the antioxidant capacity of flavonoids is predominantly positively correlated with the number of hydroxyl groups; however, the location of the hydroxyl groups may also play a role. Kaempferol contains four hydroxyl groups, two of them located on ring A and one on each of the rings B and C, respectively, which is an essential feature for good radical scavenging activity.

The decay of the ABTS radical cation was measured at 734 nm, over a period of 15 min, and used to calculate the radical scavenging capacity of free kaempferol and its copper complexes using the following Equation (1):% of ABTS radical scavenging = [1 − Abs_t_/Abs_c_] × 100(1)
where, Abs_t_ is the absorbance of kaempferol at a given time of the analysis, and Abs_c_ is the absorbance of kaempferol at time zero. The calculated values for kaempferol, kaempferol-Cu(1:1) and kaempferol-Cu(1:2) complexes are 26%, 33%, and 52%, respectively. This suggests a good ROS scavenging activity for kaempferol, which was further enhanced by the presence of Cu(II) ions. The more effective radical scavenging activity of Cu-kaempferol complexes, in comparison with the free kaempferol, can be attributed to destabilization of the π-electron system of kaempferol upon its coordination to a Cu(II) ion [32]. 

### 2.4. Protective Role of Kaempferol in Cu Fenton Raection: EPR Spin Trapping Experiment

The protective role of kaempferol in a Cu-catalyzed Fenton reaction system was studied by means of an EPR spin trapping technique, using DMPO as the spin trap, thus allowing both the nature and concentrations of ROS formed in both free Cu- and Cu-kaempferol catalyzed Fenton reactions to be determined [33]
Cu(I) + H_2_O_2_ → Cu(II) + ^•^OH + OH^−^

EPR spectra were recorded for the system containing either free copper ions, or copper ions complexed with kaempferol, and hydrogen peroxide. The anticipated formation of hydroxyl radicals in such a Fenton system was confirmed using EPR spectroscopy, in the presence of DMPO spin trap, by the observation of DMPO-OH^•^ spin adducts. 

Upon the addition of hydrogen peroxide to the control system, which contained only CuCl_2_ in a mixed solvent DMSO–H_2_O (1:4; *v*:*v*) under an air atmosphere, in the presence of the DMPO spin trap, a four line EPR signal characteristic of the ^•^DMPO-OH spin adduct (*a*_N_ = 1.486 mT, *a*_H_^β^ = 1.438 mT; g = 2.0057) was observed, as is shown in Figure 5a [34]. The addition of a small amount of kaempferol to the system (CuCl_2_–kaempferol, 5:1) led to a significant decrease in the EPR signal intensity as is shown in Figure 5a. A yet greater suppression of the free radical generation was evident when an equimolar ratio of CuCl_2_ and kaempferol was used. In addition to the ^•^DMPO-OH spin adduct, the EPR spectra obtained also revealed low intensity EPR signals which, based on a simulation analysis were attributed to superoxide radical anions, detected as the ^•^DMPO-O_2_^–^ spin adduct (*a*_N_ = 1.464 mT, *a*_H_^β^ = 1.138 mT, *a*_H_^γ^ = 0.193 mT; g = 2.0057), and methyl radicals, detected as the ^•^DMPO-CH_3_ spin adduct (*a*_N_ = 1.558 mT, *a*_H_^β^ = 2.232 mT; g = 2.0056). The formation of several different kinds of ROS provides evidence for the complex reactions that occur when Cu(II) ions interact with hydrogen peroxide [35,36] and the consecutive reactions of the reactive radicals generated with the solvent molecule (rate of the reaction between dimethyl sulfoxide and hydroxyl radicals 6.6 × 10^9^ dm^3^ mol^–1^ s^–1^ [37,38]. 

By double integration of the EPR spectra, information related to the concentrations of the ROS formed could be obtained. Thus, double integrated intensities of all three EPR spectra, as presented in Figure 5a, are shown in Figure 5b. The data show that Cu(II) chelated by kaempferol exhibits a suppressed catalytic activity in the Fenton reaction, and this results in a significantly decreased level of hydroxyl radical formation. The diminished catalytic activity of Cu(II) due to the coordinated flavonoid is in agreement with previous studies in which the catalytic activity of redox active metal ions has been linked to the availability of free metal binding sites [39,40,41,42,43].

### 2.5. Viscometric Study of the Interaction between Kaempferol/Cu-Kaempferol Complex with DNA

Viscometric measurements were made, in order to investigate the mechanism(s) of the interaction of kaempferol and Cu(II)-kaempferol complexes with DNA. The determination of the viscosity of substances provides unambiguous information about their ability and manner to interact with DNA. The interaction of a substance with DNA is associated with a change in the length of the DNA strand and is manifested as a change in the viscosity of the solution. While truncation of the DNA strand, caused by helix bending or twisting, induces a decrease in viscosity and indicates partial and/or non-classical intercalation, an increase in the length of the strand, and hence in the viscosity, is typical of classical intercalation [44,45]. The intercalating species is inserted between the nitrogen base pairs, which thus elongates the DNA molecule [46,47]. 

By comparing the viscosity values of kaempferol alone and as the Cu (II)-kaempferol complex, it was deduced that the Cu(II)-kaempferol complex had a greater intercalating ability than does free kaempferol (Figure 6). This effect is similar to that noted previously for synthetic copper complexes, with rigid fused aromatic rings such as phenanthroline and bipyridine, which often show a higher affinity for DNA than do the free ligands [48]. 

Some metallodrugs containing planar moieties may exhibit weak/mild nuclease activity which has been found to be essential for the anticancer and/or antimicrobial activity of these substances, even in the presence of hydrogen peroxide (Fenton system). From the studies published to date, as well as from our own results, emerges the potential importance of an increased dietary flavonoid intake to human health. In healthy cells, copper is present in the nucleus of DNA at physiological concentrations, and when an excess of flavonoids is present, a weak/mild intercalation of free flavonoid molecules or their Cu(II) complexes into DNA is expected. Mild intercalation of the free flavonoid molecule, or its metal complex, may produce an antioxidant effect and protect DNA from damage, by quenching the ROS at the site where the flavonoid binds to DNA [49]. 

### 2.6. DNA Damage/Protection Study Induced by Cu-Kaempferol Complex in Fenton Reaction

In the EPR spin trapping experiment described above, a protective role of kaempferol on the Cu-catalyzed Fenton reaction by means of modulation of the ROS formation was identified. However, the formation of ROS is not the only source of damage to biomolecules. In fact, Cu- or, more generally, any metal-flavonoid complexes may additionally exhibit DNA intercalating properties, as is described in the preceding section [50]. Accordingly, we studied the ROS-induced DNA damaging properties of Cu-kaempferol complexes in the presence of hydrogen peroxide (Cu-Fenton system) by means of gel electrophoresis. The effect of Cu(II)-chelation by kaempferol as a chelating flavonoid on DNA damage in the presence of hydrogen peroxide was investigated under the same experimental conditions and the resulting electrophoretic profiles are shown in Figure 7.

At low concentrations of kaempferol (Lanes 1–3), a prooxidant effect was observed, which is manifested by a significant weakening of the band from the native SC form of pDNA (p) and a slight increase in the band intensity for both LIN and OC forms of pDNA, as compared with the control (K) (Figure 7, lanes 1 and 2). DNA damage (strongly prevailing OC form) in the control experiment (see Lane K, Cu-Fenton reaction) can be attributed exclusively to the effect of ROS. Interestingly, in addition to the appearance of the DNA OC form, a slight increase in the LIN form of DNA was observed for 1:1 and 1:2 Cu-kaempferol complexes (see Lanes 1 and 2). This can be explained by a moderate intercalating effect of the Cu-kaempferol complexes, as was confirmed by viscometric experiments. Here, it should be noted that while EPR spin trapping experiments confirmed a suppressed formation of ROS upon coordination of kaempferol to Cu(II) ions, gel electrophoresis was not sufficiently sensitive to show a difference between DNA damage caused by Cu(II)-Fenton and Cu(II)-kaempferol Fenton systems. Increasing the concentration of kaempferol (Figure 7, Lanes 3–8) resulted in a DNA protecting effect, which was observed in the gel as a gradual weakening, then complete loss of the linear band (LIN) of pDNA, and an increase in the band intensity from the native form of pDNA. The results indicate that the effect of kaempferol is similar to that of quercetin [4]. At its lower concentrations, kaempferol had no protective effect on DNA (observed in terms of relative DNA damage), whereas when kaempferol was present in excess, a DNA-protective effect was demonstrated.

It appears likely that the comparable behavior of both flavonoids, quercetin and kaempferol, is related to the structural similarity of their molecules at those positions designed to be copper interaction sites, i.e., the 3-hydroxyl and 4-oxo groups on the C-ring (Figure 1).

### 2.7. ROS Involved in DNA Damage by Cu-Kamferol Fenton System

In order to identify the nature of ROS formed in the Cu-catalyzed Fenton reaction, the following ROS probes were used: L-histidine (a scavenger of singlet oxygen) DMSO (a scavenger of hydroxyl radicals) and the SOD enzyme (a superoxide radical anion scavenger) [51].

To verify the effectiveness of these standard ROS scavengers, a reaction mixture was studied containing plasmid DNA, along with copper chloride and hydrogen peroxide (Cu-Fenton system), in the total absence of any flavonoids. In the corresponding electrophoretic profile (Figure 8), a significant conversion of the native SC form of pDNA can be seen, in Lane 1, as a result of attack by ROS formed in the Cu-Fenton reaction. Following the addition of specific ROS scavengers (L-histidine, DMSO, and SOD), the clear preservation of the SC form of pDNA indicates the formation of singlet oxygen, hydroxyl radicals, and superoxide radical anions, respectively (Figure 8). 

The formation of ROS in the Cu-Fenton-like reaction was studied in the presence of kaempferol at two selected concentrations of 10 and 200 μM (Figure 9, Lanes 1). The effects of adding the ROS scavengers, L-Histidine, DMSO, and SOD, used for the detection of singlet oxygen, hydroxyl radicals, and superoxide radical anions, respectively (lanes 2–4), were determined from the intensity of the bands pertaining to the particular conformational states of pDNA. 

From the electropherogram (Figure 9), it can be seen that at lower concentrations of kaempferol (10 μM), the combined effect of all three ROS, singlet oxygen, hydroxyl radicals, and superoxide radical anions still caused significant damage to the DNA (Figure 9, Lane 1, left panel). However, at a significantly higher concentration of kaempferol (200 μM) some partial protection of the DNA from damage mediated by all three types of ROS was conferred (Lane 1, right panel). Following the application of either the singlet oxygen scavenger (L-histidine, Lane 2) or the superoxide radical anion scavenger (SOD, Lane 4), the observed DNA damage was relatively minor. Thus, it may be deduced that the combined effect of hydroxyl radicals and superoxide radical anions, or of hydroxyl radicals and singlet oxygen, does not cause a significant level of DNA damage. On the other hand, the application of a hydroxyl radical scavenger (DMSO, Lane 3) indicated that the combined effect of singlet oxygen and superoxide radical anions causes considerable DNA damage. The level of damage in the presence of DMSO could be further enhanced by the action of methyl radicals, which are generated from hydroxyl radicals according to the reaction [52]:
CH_3_SOCH_3_ + ^•^OH → ^•^CH_3_ + CH_3_SOOH

These results also imply, in agreement with previous studies, that kaempferol is a good scavenger of hydroxyl radicals, even though its capacity to alleviate DNA damage caused by singlet oxygen and superoxide radical anions is limited [53]. We note that in the Fenton system studied, in using DMPO as the spin trap, we were able to detect predominantly hydroxyl radicals and also superoxide radical anions and methyl radicals, though in significantly smaller amounts. However, detection of ROS using the EPR spin trapping technique has some analytical limitations. For example, the reaction of DMPO with the superoxide radical anion is rather slow, and its determination is also affected by the presence of cupric ions, meaning that the amount of superoxide radical anion detected may not correspond to the total concentration of this species formed under the given experimental conditions. 

## 3. Materials and Methods

### 3.1. Materials

All chemicals and solvents used were of analytical grade. Copper chloride, thymus DNA (CT-DNA, calf thymus) and the kaempferol flavonoid were purchased from Merck (Darmstadt, Germany), and bromphenol blue from AppliChem GmbH (Darmstadt, Germany). Dimethyl sulfoxide and glycerol were obtained from Merck (Darmstadt, Germany), hydrogen peroxide from Mikrochem (Pezinok, Slovakia), and tryptone and agar from Merck (Darmstadt, Germany). ABTS ((2,2-azino-bis(3-ethylbenzothiazoline-6-sulphonic acid) was obtained from Merck (Darmstadt, Germany). The 1 kb DNA ladder was from Solis BioDyne (Tartu, Estonia), ethidium bromide (EtBr) was from Serva (Heidelberg, Germany), EcoR1 restriction was from Thermoscientific (Rockford IL, USA), and the Isolate II Plasmid Kit from Bioline (Memphis TN, USA). Tris-NaCl buffer (5 mM TRIS, 50 mM NaCl, pH 7.2) was from Aniara Diagnostica (West Chester OH, USA). 

### 3.2. EPR Spectroscopy

EPR spectra of copper chloride and Cu-flavonoid complexes (1:1 and 1:2 ratios) were measured in frozen DMSO solution (77 K) using quartz cylindrical cells on a Bruker EMX Plus spectrometer at X-band (9.4 GHz) [54]. EPR data were simulated using commercially available software SimFonia, version 1.2 (Bruker, Billerica, MA, USA).

### 3.3. UV-Vis Spectroscopy

UV Vis spectra were recorded using spectrophotometer Shimadzu in 3 mL quartz cuvettes at room temperature. At first, spectra of kaempferol were measured as a DMSO and methanol solution (c = 0.1 mM). Then, solution of kaempferol was mixed with solution of CuCl_2_ (DMSO or methanol) using molar ratio Cu–kaempferol = 1:2, or 1:1 respectively. Spectra were recorded 5 min after mixing.

### 3.4. Radical Svavenging Activity of Kamepferol and Its Cu(II) Complexes

An ABTS (2,2′-azino-bis(3-ethylbenzothiazoline-6-sulphonic acid) assay was used to evaluate the radical scavenging activity of kaempferol or Cu-kaempferol complexes. A solution of the ABTS radical cation was prepared by dissolving 17.2 mg of ABTS and 3.3 mg of K_2_S_2_O_8_ in 5 mL of deionized water and leaving this in dark for 24 h to ensure full oxidation of the ABTS salt. A stock solution of the ABTS radical cation was prepared by diluting 1 mL of the oxidized solution with 60 mL of deionized water. 1 mM solutions of both kaempferol and CuCl_2_ were prepared in DMSO. The time decay of the ABTS^•+^ signal at 734 nm in the presence of kaempferol or its Cu complexes (Cu–kaempferol = 1:1 and 1:2) was monitored during 900 s, using a UV Vis NIR spectrometer (Shimadzu), until a steady state level of the absorbance signal was achieved [55]. 

### 3.5. EPR Spin Trapping Experiment

The generation of reactive radical species was monitored using the EMX Plus X-band EPR spectrometer with a High Sensitivity Probe-head (Bruker) and the small quartz flat cell (Wilmad-LabGlass, model WG 808-Q). The EPR spin trapping measurements were performed using 5,5-dimethyl-1-pyrroline *N*-oxide (DMPO, Sigma-Aldrich; distilled prior to the application) as the spin trapping agent. The reference Fenton system contained CuCl_2_ and hydrogen peroxide in a mixed water/DMSO (4:1; *v*:*v*) solvent under an air atmosphere. Kaempferol was dissolved in DMSO (Merck, SeccoSolv) and mixed with aqueous CuCl_2_ in different ratios (Cu–Kaempferol, 5:1 and 1:1) and with DMPO added. An aqueous solution of hydrogen peroxide was added to the mixture, to initiate the Fenton reaction and the EPR spectra were recorded 2 min after the H_2_O_2_ addition. The concentrations of spin-adducts were evaluated by a double integration of the recorded EPR spectra, in reference to a calibration curve obtained from the EPR spectra of Tempol (4-hydroxy-2,2,6,6-tetramethylpiperidine *N*-oxyl; Sigma-Aldrich) solutions, which were measured under strictly identical experimental conditions. All experiments were carried out in triplicate. The experimental EPR spectra of the spin adducts were analyzed and simulated using the SimFonia, version 1.2 (Bruker, Billerica, MA USA) [56] and EasySpin, version 5.2.28 simulation toolbox working within Matlab software (Natick MA, USA) [57].

### 3.6. Viscosity Measurements

The interaction of flavonoids and Cu(II)-flavonoid complexes with DNA was studied by means of viscosity measurements. The mixture contained CT-DNA (1 mM) in Tris-NaCl buffer (5 mM TRIS, 50 mM NaCl, pH 7.2) along with the flavonoid/Cu-flavonoid complex. The temperature of this mixture was 25 ± 0.1 °C and the ratio (flavonoid/Cu-flavonoid complex): DNA was 0.2, 0.4, 0.6, 0.8, and 1.0. The data were presented as the (η/η_0_)^1/3^ vs. (flavonoid/Cu-flavonoid complex) ratio, where η is the viscosity of DNA in the presence of the flavonoid or Cu-flavonoid complex, and η_0_ is the viscosity of DNA in the buffer medium.

### 3.7. DNA Damage Study

#### 3.7.1. Isolation of Plasmid DNA 

Plasmid DNA pBSK+ was purified using a Plasmid Mini-Prep Kit from a liquid culture of *E. coli* DH5α bacterial cells grown in LB medium containing Ampicilin (50 μg·mL^−1^) at 37 °C for 20 h under 160 rpm shaking. Plasmid DNA was eluted in phosphate buffer (50 mM, pH 7.2). The purity and concentration of DNA were evaluated using a spectrophotometer NanoDrop (USA). The DNA purity was determined from the absorbance ratios A_260_/A_230_ (2.0–2.2) and A_260_/A_280_ (≥1.8) which confirmed it to be sufficient for the work described.

#### 3.7.2. Analysis of DNA Damage by Gel-Electrophoresis

DNA damage induced by flavonoids or Cu-flavonoid complexes (5 μM CuCl_2_ and varying concentrations of flavonoids) was evaluated from the transformation of plasmid to nicked and linear DNA. Flavonoids/Cu-flavonoid complexes dissolved in DMSO were left to stand for 10 min, and into this solution was added 15 μM solution of plasmid DNA in phosphate buffer at pH 7.2. To initiate the Cu-catalyzed Fenton reaction, a 30% solution of hydrogen peroxide is water was added, and the DNA was analyzed after 30, 60, and 120 min periods. 

Into 15 μL of the studied sample was added 6 μL of quench buffer consisting of 0.25% bromphenol blue and 30% glycerol. The samples were subjected to an electrophoretic analysis for 1.6 h at 80 V using 0.8% agarose gel in TBE buffer (pH = 8.0) containing 0.5 μg/mL of EtBr. As a standard, the 1 kb DNA ladder was used, and a plasmid sample linearized with EcoRI endonuclease was employed as the control for double strand breaks. Each experiment was performed in triplicate, control experiments were performed with cupric ions and flavonoids, both at 5 μM concentrations, and the gels were imaged under ultraviolet light [58]. 

#### 3.7.3. Identification of ROS

In order to characterize the nature of radical species (ROS) formed in the course of Cu-flavonoid catalyzed Fenton reactions, the following radical scavengers were used: L-histidine (20 mM) as a scavenger for singlet oxygen (^1^O_2_), DMSO (6 µL) as a scavenger for hydroxyl radicals (^•^OH), and the superoxide dismutase (SOD) enzyme (15U) as a scavenger of superoxide radical anions (O_2_^•−^) [51]. 

## 4. Conclusions

Two currently very widespread disease states of living organisms, cancer and neurological diseases, can be characterized by opposing mechanisms. While neurological conditions such as Alzheimer’s disease are characterized by premature cell death, cancer is characterized by an enhanced resistance to cell death. However, oxidative stress appears to be a common denominator in both these disease states, which can be considered to arise from an imbalance between antioxidant and prooxidant mechanisms. Thus, the testing of bifunctional drugs that display antioxidant properties under certain conditions, but prooxidant properties under others, is of vital importance. Flavonoids exhibit both antioxidant and prooxidant properties depending on, e.g., the presence of redox active metals such as copper(II). Accordingly, both antioxidant and prooxidant properties of kaempferol were investigated. Prooxidant conditions were simulated using a Cu-catalysed Fenton reaction, motivated by the fact that a disturbed metabolism of copper is typical both for Alzheimer’s disease and certain types of cancers. 

The EPR spectroscopy of Cu–kaempferol (1:1 and 1:2) complexes in DMSO confirmed interaction between Cu(II) and kaempferol and the existence of two Cu-kaempferol species in frozen solution which corresponded to various coordination environments around copper(II) ions. A more precise information on the Cu-kaempferol interaction was obtained from UV-vis spectra which confirmed the interaction of the 5-OH group on ring A and the 4-CO group on ring C of kaempferol with Cu(II) ions in DMSO solution. On the contrary, the UV-vis spectra in methanol revealed the presence of a tight interaction between the 4-CO and 3-OH groups of the kaempferol ring C and the Cu(II) ion. The discrepancy of the results can be explained in terms of the greater strength of the interaction between DMSO molecules, than exists for methanol molecules with Cu(II) ions. 

Kaempferol and, even more so, its copper complex exhibit effective radical scavenging activities according to ABTS assay. The latter radical scavenging enhancement can be explained in terms of a perturbation of the π-electron system that occurs when kaempferol coordinates to a Cu(II) ion, which leads to a more facile oxidation of the flavonoid moiety.

The protective role of kaempferol in a copper catalyzed Fenton type reaction system has been studied using an EPR spin trapping technique. The results demonstrated that coordination of kaempferol to Cu(II) ions results in a suppressed formation of hydroxyl radicals by 80%. This can be explained by a diminished catalytic activity of Cu(II) due to it being coordinated to the flavonoid, and is broadly in agreement with previous studies in which the catalytic activity of redox active metal ions has been linked to the availability of free metal binding sites.

As expected, viscometric studies revealed that the Cu(II)-kaempferol complex exhibits a greater intercalating ability than does free kaempferol. These results were compared with those from the DNA damage study of Cu-kaempferol complexes in a Fenton reaction system using gel electrophoresis, where it was found that, at low concentrations of kaempferol (Cu–kaempferol ratios 1:1 and 1:2), a weak protective effect against DNA damage was obtained. Also, at such low concentrations of kaempferol, and in agreement with the viscometric measurements, a weak band corresponding to linear DNA (LIN) was observed, which can be related to the weak intercalating ability of Cu-kaempferol complexes. Increasing the concentration of kaempferol resulted in a DNA protecting effect, which was observed in the gel as a gradual weakening and then complete loss of the linear band (LIN) of pDNA, with an increase in the band intensity from the native form of pDNA. At the lowest concentrations, kaempferol showed almost no protective effect against DNA damage, whereas, when it was present in excess, some protection was found. This can be explained by the presence of kaempferol molecules at the surface of DNA, providing protection against attack by ROS which originate from the Fenton reaction involving intercalating Cu(II)-kaempferol complexes. 

Application of the ROS scavengers, L-histidine, DMSO, and SOD, confirmed the formation of singlet oxygen, hydroxyl radicals, and superoxide radical anions, respectively. 

Based on our results, obtained under simulated Fenton conditions, we propose that the prooxidant properties of kaempferol or Cu-kaempferol complexes may provide anticancer activity of these substances, and furthermore, that its coordination to Cu(II) may enhance the bioavailability of Kaempferol. Most of the previously reported investigations into the anticancer potency of kaempferol were made using in vitro conditions, making it difficult to extrapolate to the in vivo situation, and hence to any potential clinical usefulness. As mentioned above, while kaempferol has been shown to be an effective supplement to act against various types of cancers, the most promising results have been obtained in the case of breast cancer. Thus, when kaempferol was administered in a combination with other polyphenols, a very promising anticancer activity was demonstrated, in a robust preclinical study, using MCF-7 and MDA-MB-231 breast cancer cell lines [59]. Since poor bioavailability of kaempferol is a significant obstacle, the use of kaempferol-based nanoparticles or metallodrugs, e.g., copper-kaempferol complexes, represents a promising approach for developing flavonoid anticancer strategies. More detailed experiments, aimed to determine the most effective dose of kaempferol, the optimum stage of its administration, or these factors in its combination with other flavonoids, are needed.

When present in excess, kaempferol displays antioxidant properties, under Cu-Fenton conditions. This suggests that kaempferol might prove a suitable candidate for the prevention or treatment of oxidative stress related medical conditions that involve a disturbed metabolism of redox metals such as copper, for example, Menkes disease, and neurological disorders, including Alzheimer’s disease. Kaempferol coordinated to free copper ions may reduce the level oxidative stress in the brains of Alzheimer’s patients. Previous studies have shown that the dietary administration of high doses of quercetin to patients suffering from neurological disorders, resulted in improvements to their cognitive functions [60]. Optimization of the necessary dose size and establishment of the critical age of the patient, at which to start kaempferol and other flavonoids supplementation for preventative therapy and/or treatment, are important issues for further investigation [61,62,63,64,65,66,67,68,69,70]. It is probable that such an optimized long-term preventive supplementation strategy may reduce the occurrence and delay the incidence of redox metal/oxidative stress-related diseases such as neurological disorders with disturbed metabolism of redox metals.

## Figures and Tables

**Figure 1 ijms-22-01619-f001:**
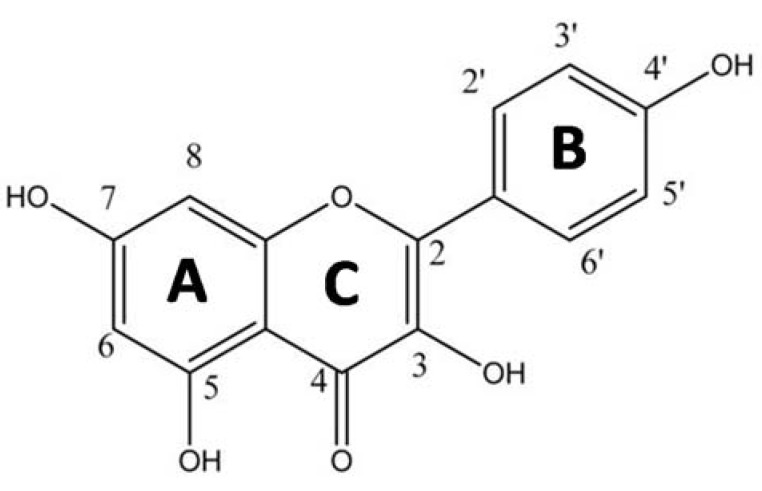
Structure of kaempferol.

**Figure 2 ijms-22-01619-f002:**
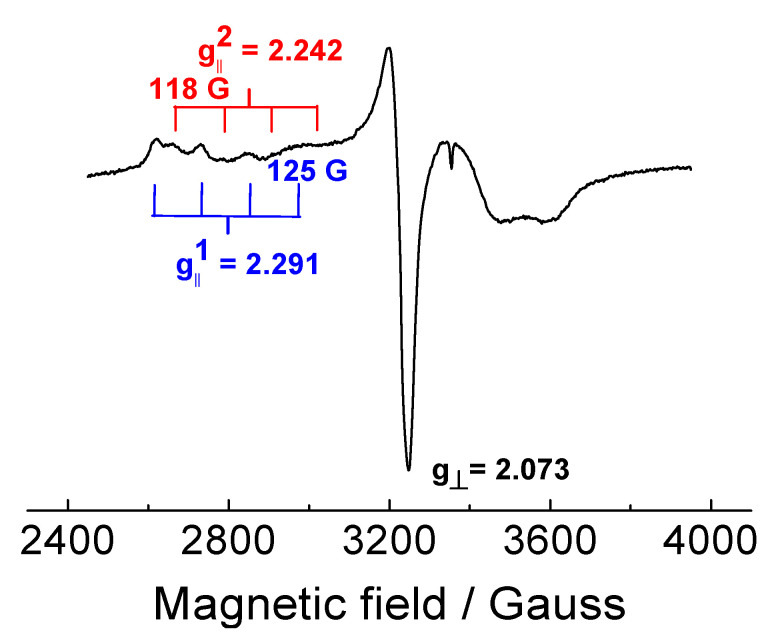
EPR spectrum of in DMSO solution at 110 K. Concentration of Cu was 1 mM.

**Figure 3 ijms-22-01619-f003:**
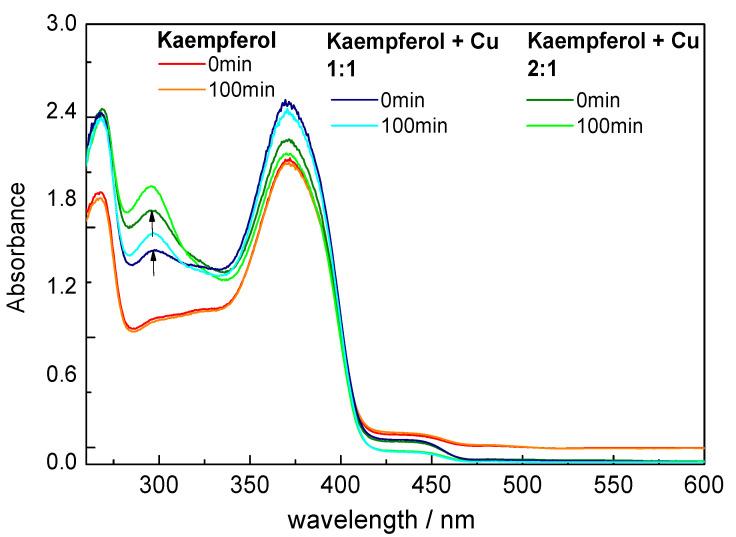
UV-vis spectra of kaempferol and its Cu(II) complexes (1:1 and 1:2) in DMSO. [Cu(II)] = 0.1 mM. The spectra were recorded 5 min after mixing of the copper solution with kaempferol.

**Figure 4 ijms-22-01619-f004:**
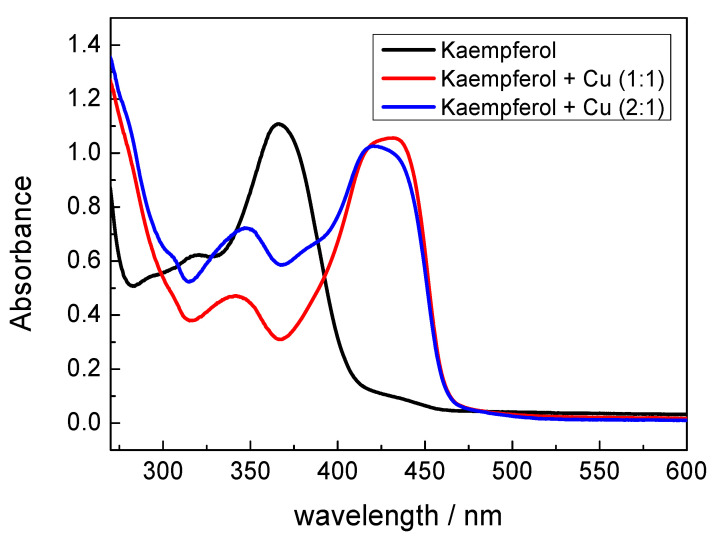
UV-vis spectra of kaempferol and its Cu(II) complexes (1:1 and 1:2) in methanol. [Cu(II)] = 0.1 mM. The spectra were recorded 5 min after mixing of the copper solution with kaempferol.

**Figure 5 ijms-22-01619-f005:**
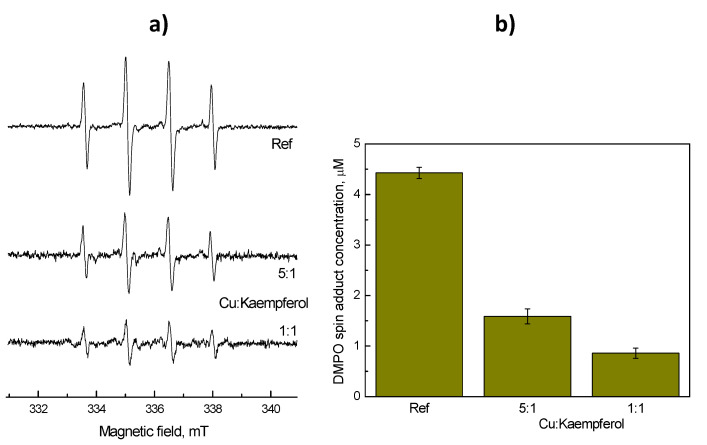
The effect of Kaempferol, present in two different Cu–Kaempferol molar ratios (5:1 and 1:1), on the formation of free radicals by means of a copper-catalyzed Fenton reaction. (**a**) Experimental EPR spectra obtained in the system CuCl_2_/H_2_O_2_/DMPO/DMSO–H_2_O (1:4; v:v) under an air atmosphere, without (Ref) or with Kaempferol. (**b**) The total concentration of DMPO spin adduct concentration monitored in DMSO/water (1:4; v:v) solution of CuCl_2_ alone (Ref) or in the presence of Kaempferol containing DMPO spin trapping agent after the addition of hydrogen peroxide. Initial concentrations: *c*(CuCl_2_) = 0.2 mM; *c*(Kaempferol) = 0/0.04/0.2 mM; *c*(DMPO) = 0.02 M; *c*(H_2_O_2_) = 0.01 M. EPR parameters: ^•^DMPO-OH (*a*_N_ = 1.486 mT, *a*_H_^β^ = 1.438 mT; g = 2.0057); ^•^DMPO-O_2_^–^ (*a*_N_ = 1.464 mT, *a*_H_^β^ = 1.138 mT, *a*_H_^γ^ = 0.193 mT; g = 2.0057); ^•^DMPO-CH_3_ (*a*_N_ = 1.558 mT, *a*_H_^β^ = 2.232 mT; g = 2.0056).

**Figure 6 ijms-22-01619-f006:**
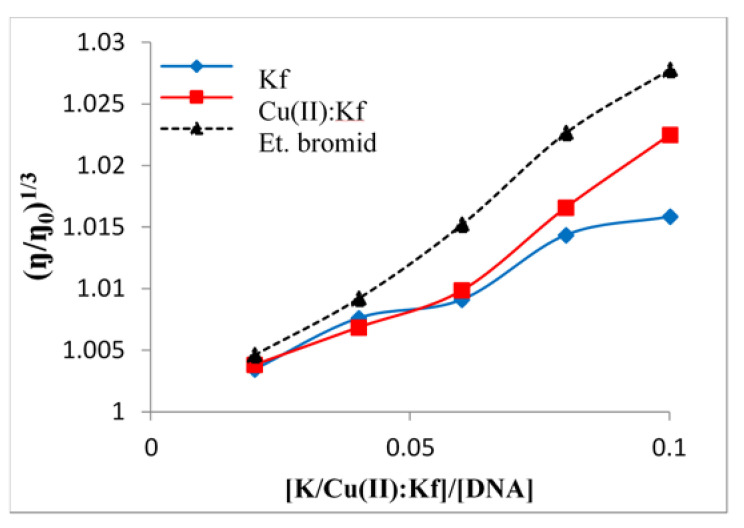
Relative viscosity of CT-DNA in buffer (5 mM TRIS, 50 mM NaCl, pH 7.2) influenced by increasing concentration of free kaempferol and Cu-kaempferol complex at 25 ± 0.1 °C compared to ethidium bromide (EtBr).

**Figure 7 ijms-22-01619-f007:**
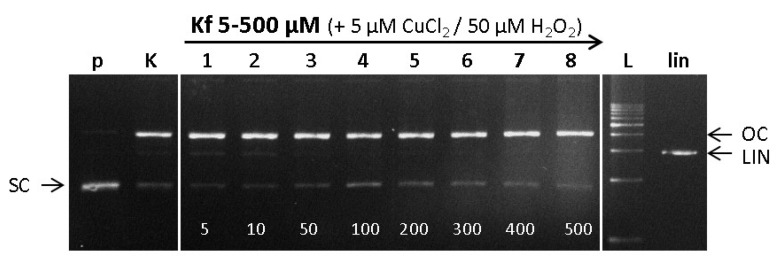
Electrophoretic profile of agarose gel (0.8%), 15 µM plasmid PBSK + DNA with Cu(II) ions, hydrogen peroxide and kaempferol incubated for 30 min. Samples were loaded in the gel lanes in the following order: (p) pDNA–control; (K) pDNA + 5 µM CuCl_2_ + 50 µM H_2_O_2_-control Cu-Fenton reaction; (1–8) pDNA + 5 µM CuCl_2_ + 50 µM H_2_O_2_ + (1) 5 µM kaempferol, (2) 10 µM kaempferol, (3) 50 µM kaempferol, (4) 100 µM kaempferol, (5) 200 µM kaempferol, (6) 300 µM kaempferol, (7) 400 µM kaempferol, (8) 500 µM kaempferol; (L) 1 kb DNA standard, (lin) pDNA linearized with EcoR1 endonuclease.

**Figure 8 ijms-22-01619-f008:**
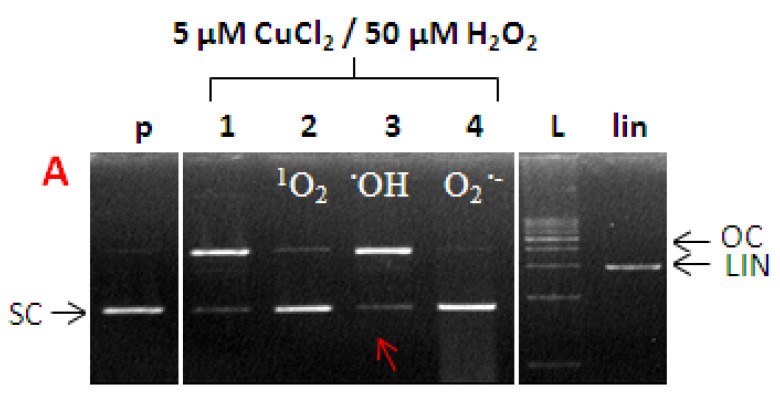
Electrophoretic profile of agarose gel (0.8%) containing 15 µmol plasmid DNA pBSK+ with Cu (II) ions and hydrogen peroxide. Samples were loaded in the gel lanes in the following order: (p) pDNA-control; (1) pDNA + 5 µM CuCl_2_, 50 µM H_2_O_2_, (2–4) pDNA + 5 µM CuCl_2_ + 50 µM H_2_O_2_ plus (2) L-histidine, (3) DMSO, (4) SOD. (L) 1 kb DNA standard, (lin) pDNA linearized with EcoR1 endonuclease.

**Figure 9 ijms-22-01619-f009:**
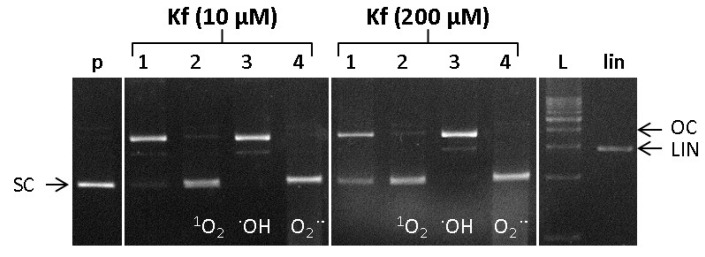
Electrophoretic profiles of agarose gel (0.8%) containing 15 µmol plasmid DNA pBSK+ with Cu(II) ions and hydrogen peroxide, to which 10 µmol and 200 µmol of kaempferol had been added, respectively. Samples were loaded in the gel lanes in the following order: (p) pDNA-control; Left panel: (1) pDNA + 5 µM CuCl_2_ + 50 µM H_2_O_2_ + 10 µM kaempferol, (2) pDNA + 5 µM CuCl_2_ + 50 µM H_2_O_2_ + 10 µM kaempferol + L-histidine; (3) pDNA + 5 µM CuCl_2_ + 50 µM H_2_O_2_ + 10 µM kaempferol + DMSO; (4) pDNA + 5 µM CuCl_2_ + 50 µM H_2_O_2_ + 10 µM kaempferol + SOD; Right panel: (1) pDNA + 5 µM CuCl_2_ + 50 µM H_2_O_2_ + 200 µM kaempferol; (2) pDNA + 5 µM CuCl_2_ + 50 µM H_2_O_2_ + 200 µM kaempferol + L-histidine; (3) pDNA + 5 µM CuCl_2_ + 50 µM H_2_O_2_ + 200 µM kaempferol + DMSO; (4) pDNA + 5 µM CuCl_2_ + 50 µM H_2_O_2_ + 200 µM kaempferol + SOD. (L) 1 kb DNA standard, (lin) pDNA linearized with EcoR1 endonuclease, (µM = µmol.dm^−3^).

**Table 1 ijms-22-01619-t001:** EPR data of Cu–kaempferol complex (1:1) in DMSO at 100 K.

System	g_⊥_	g_‖_ ^a^	A_‖_ (Gauss) ^a^
Cu–kaempferol	2.073	2.291	125
		2.242	118

^a^ Two distinct copper sites have been detected in the parallel region of the spectrum.

## Data Availability

Not applicable.

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
