# Peer review of "Antioxidant vs. Prooxidant Properties of the Flavonoid, Kaempferol, in the Presence of Cu(II) Ions: A ROS-Scavenging Activity, Fenton Reaction and DNA Damage Study"

_ijms, 2021, doi:10.3390/ijms22041619_

Round 1

Reviewer 1 Report

The original contribution by Simunkova and coauthors reports on the antioxidant and prooxidant properties of kaempferol. A particular attention is devoted to the role of Cuii ions affecting antioxidant properties of kaempferol. Paper contains several interesting and important in vitro results with the relevance for further potential use of kaempferol in clinics and further in vivo testing. 

Comments/suggestions:

  1. The epr data for cu chelates with kaempferol is shown in Fig. 2. I would recommend authors to summarize the epr data in a Table. 
  2. Authors may briefly discuss the biological impact of ROS probed by three probes with the conclusions obtained by epr spin trapping. 
  3. In the section "Conclusions" authors may supplement the results with the known epidemiological results. 

Overall summary: This is a solid paper which bridges the gap between instrumental studies on flavonoids and in vivo testing for prospective use of flavonoids in clinics. 

Reviewer 2 Report

The paper is sound and is providing additional information and experimental proofs on the well-recognized antioxidant activity of kaempferol and its cupric complexes.

The amount of data provided is high, some of the experiments and colclusions being sound and positive impacting the knowledge in the field.

I reccomend the paper publication after small improvement of the part concerning ABTS assay:

-please underline the significance of the ABTS test/assay for investiganted flavonoid. Suggestion- please move figure 5 to supplemental material and introduce instead the equation describing the kinetics of the reaction using these infromation as basis to underline the significance of the ABTS test and the correlation between the kaempferol and its complexes behaviour  with concentration level

Please restructure the conclusion part to emphasize better the achieving of the paper aim and the potential of the published results to contibute to human treatment with flavonoids suplements.

Please pay attention to the typing errors (e.g 3.7.2 paragraph title, page 16)

Reviewer 3 Report

This manuscript evaluated the effect of kaempferol on DNA damage. In general, the experimental methods and results are nice. However, it is hard to find new findings in this study. In authors' previous study, almost same and more detailed results were published (Ref. 4). Then, what could be the new findings in this study?

Round 2

Reviewer 3 Report

I think the answers for my question are now included in result and discussion section.

This manuscript is now acceptable.